# RhAGL24 Regulating Auxin-Related Gene *RhARF18* Affects Stamen Petaloidy in Rose

Lin Liu, Yanchao Guo, Zhicheng Wu, Haoran Ren, Yunhe Jiang, Nan Ma, Junping Gao and Xiaoming Sun * 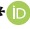

Beijing Key Laboratory of Development and Quality Control of Ornamental Crops, College of Horticulture, China Agricultural University, Beijing 100193, China; s20193172480@cau.edu.cn (L.L.); bhsgych@163.com (Y.G.); wuzc@cau.edu.cn (Z.W.); rhr18800175823@126.com (H.R.); yunhe.jiang@cau.edu.cn (Y.J.); ma_nan@cau.edu.cn (N.M.); gaojp@cau.edu.cn (J.G.)
* Correspondence: xmsun@cau.edu.cn; Tel.: +86-10-62733848

**Abstract:** AGAMOUS-LIKE 24 (AGL24) is a key gene regulating floral transition, but its involvement in flower organ identity remains largely unknown. In this study, we found that *RhAGL24* is strongly related to petal and stamen development in rose. Its expression increases rapidly at the petal primordium development stage and maintains a high level until the complete differentiation stage. *RhAGL24* silencing increases the number of malformed petals and decreases the number of stamens, indicating that this gene affects stamen petaloidy. *RhAG* (*AGAMOUS*), a class C gene associated with petal and stamen development, is downregulated in *RhAGL24*-silenced plants. Moreover, we found that RhAGL24 could directly bind to the promoter region of *RhARF18* (*AUXIN RESPONSE FACTORS 18*), a regulator of *RhAG*. Our results suggested that RhAGL24-*RhARF18* module regulates stamen petaloidy in rose and provide new insights into the function of *AGL24* for plants.

**Keywords:** *Rosa hybrida*; *RhAGL24*; stamen petaloidy; transcriptional regulation; *RhARF18*

## 1. Introduction

Stamen petaloidy occurs in roses, and the number of petals and stamens is related to ornamental value. Stamen petaloidy increases the diversity and appreciation of rose varieties. Studies on stamen petaloidy have great important biological significance and industrial value. At present, many reports about flower organ development are available, but studies on the mechanism of stamen petaloidy in roses are rare.

Floral organ development has always been a hot topic in plant research. According to the familiar ABCDE model, flowers have four whorls, namely, sepals, petals, stamens, and carpels arranged from the outside to the inside [1]. Four genes, namely, *AGAMOUS* (*AG*), *APETALA2* (*AP2*), *APETALA3* (*AP3*), and *PISTILLATA* (*PI*), can affect floral organ development but not the location of primordia and are expressed at different locations in cells, resulting in different states of differentiation. The functions of these genes also have overlapping fields in the concentric circles of flower primordia [2]. *AP2* plays a role in the two outer whorls, *PI* and *AP3* are involved in the second and third whorls of development, and *AG* functions in the two inner whorls. The combined gene products determine the four organ types of wild-type flowers. The products of *AG* and *AP2* genes have antagonistic effects. AP2 inhibits *AG* gene expression in the outer two whorls and *AG* inhibits *AP2* gene expression in the inner whorls [3]. SEUSS (SEU) and LEUNIG (LUG) negatively regulate *AG* gene expression in the first two whorls of flowers [4]. *lug ag* double mutations have the same phenotype as *ag* single mutations, and *AG* is epistatic to *LUG* [5]. Stamen-specific patterns are dependent on *LUG* but not on *AP2* [6]. *AP3* and *PI* play a redundant role in floral meristem formation [7]. AP3/PI interacts with other factors in early flower development to directly inhibit *AP1* expression [8].

In addition, floral organ development is mainly related to auxin. Auxin plays a key role in flower primordium development [9]. The auxin transport protein PIN-FORMED1

(PIN1) is a transmembrane protein with polar localization and participates in auxin efflux. The inflorescence of the mutant *pin1* usually has no flower [10,11]. However, exogenous auxin application on inflorescence of tomato *pin1* can induce the formation of floral primordia [12]. This suggests that auxin is required for normal development of floral primordia. The phenotypes of *pinoid* (*pid*) and *monopteros* mutants are similar to the *pin1*, and neither can produce normal flower buds [13,14]. Auxin response factors (ARFs) are a family of transcription factors that specifically bind to auxin elements to regulate downstream gene expression [15,16]. The auxin response factor family has many members with different functions [17,18]. For example, *ETTIN/ARF3* is involved in floral meristem determinism [19], and the mutation of *ETTIN/ARF3* can enhance the phenotype of *ag* mutants [20,21]. *ARF4* regulates flowering time [22]. *MONOPTEROS/ARF5* promotes bud formation [23] and participates in the development of male and female gametes [24]. *ARF18* inhibits auxin transduction [25], regulates angular wall development [26], affects petal and stamen development, and negatively regulates *RhAG* expression [27]. Significant differences in auxin-related gene expression are also found between the single and double flower transcriptomes of apple species [28].

The flowering pathway is a complex regulatory network that is influenced by various factors. *FLOWERING LOCUS T* (*FT*), *SUPPRESSOR OF OVEREXPRESSION OF CO1* (*SOC1*), and *LEAFY* (*LFY*) are three important regulatory factors in *Arabidopsis thaliana* [29,30]. Located in inflorescence and floral meristems, *AGL24* mediates the communication among these three genes and is activated during floral transition. *AGL24* is a transcription factor that dose-dependently promotes flowering, and its overexpression leads to early flowering [31]. *AGL24* and *SOC1* have similar functions of stimulating flowering and are upregulated through vernalization or by gibberellin to form a positive feedback loop [29]. By contrast, *AGL24* mRNA levels are not affected by *FLC*, so *AGL24* is a *FLC*-independent target in the vernalization pathway [32]. *SHORT VEGETATIVE PHASE* (*SVP*), a gene with extremely high homology to *AGL24*, exhibits completely opposite effects. Although both are important regulatory factors in flower transformation, *SVP* is the suppressive factor of flowering, and *AGL24* is the promoter of flowering [31]. In *svp agl24* double mutants, *SVP* is epistatic to *AGL24*, and the number and characteristics of flower organs are affected. Yeast hybridization experiments showed that the dimer of AP1-SVP or AP1-AGL24 could bind to LUG-SEU to inhibit *AG* activity. Moreover, the phenotype of the triple-mutant *ap1 agl24 svp* is more distinct than that of *agl24 svp*, suggesting that the three genes have functional redundancy in *AG* regulation during the early stage of flower development [33].

AGL24 has been studied as a regulator of flowering time in many species, but its relationship with floral organ identity is rarely reported. Here, we found that *RhAGL24* is highly expressed during petal and stamen primordia development. Therefore, we aimed to explore the function and regulatory mechanism of *RhAGL24* in petal and stamen development. We discovered a RhAGL24-*RhARF18* module that regulates stamen petaloidy in rose and provided new insights into the function of *AGL24* for plants.

## 2. Materials and Methods

### 2.1. Plant Materials and Growth Conditions

Sterile tissue culture seedlings of *Rosa hybrida* 'Samantha' were grown on Murashige and Skoog (MS) medium containing plant hormones [34]. Upon growing stems and leaves, the tissue culture seedlings were supplemented with the hormone ratio of 1.0 mg L$^{-1}$ 6-benzylaminopurine (6-BA), 0.01 mg L$^{-1}$ α-naphthalene acetic acid (NAA), and 3 mg L$^{-1}$ gibberellin acid (GA3). After 1 month of growth, the specimens were transferred to the rooting medium of 1/2 MS medium supplemented with 0.1 mg L$^{-1}$ NAA. After 1 month of growth, the tissue culture seedlings took root and were transplanted into a pot with a 1:1 mixture of peat and vermiculite. *Nicotiana benthamiana* was grown under the same conditions as rooting roses. Both plants were reared at $22 \pm 1\ ^{\circ}$C under a 16 h light and 8 h dark photoperiod.

### 2.2. RNA Extraction and Quantitative RT-PCR Analysis

Total RNA of rose was extracted by thermal borate method [35]. In the reverse transcription experiment, the HiScript® II Q RT SuperMix for qPCR (+gDNA Wiper) kit (Vazyme, Nanjing, China) was used for 1 μg RNA reverse transcription into cDNA in accordance with the manufacturer's instructions. Quantitative RT-PCR (qRT-PCR) was performed with StepOnePlus™ Real-Time PCR system (Applied Biosystems, Carlsbad, CA, USA) using the KAPA SYBR FAST Universal qRT-PCR kit (Kapa Biosystems, Boston, MA, USA). The specific test steps are carried out in accordance with the instructions. Relative gene expression was calculated by the $2^{-\Delta\Delta CT}$ method [36]. *RhUBI2* gene with stable expression was selected as internal control [37], and the primers used for qRT-PCR were designed in NCBI [38] as shown in Table S1.

### 2.3. Cloning and Sequence Analysis

NCBI and Primer Premier 5.0 software were employed for primer design (Table S1) using the sequence of *RchiOBHm_Chr7g0205071* from 'Old Blush' genome as the reference [39]. The open reading frame (ORF) of *RhAGL24* was cloned from the cDNA of 'Samantha' flower bud and subsequently corrected. The promoter of *RhARF18* was cloned and corrected in the same way. The amino acid sequence of RhAGL24 was translated by an online translate website [40]. The protein conserved domains were predicted with NCBI. The *cis*-acting element was predicted by JASPAR [41], and the *cis*-acting element on the *RhARF18* promoter was detected with FIMO [42].

The protein sequences of target genes were downloaded from NCBI and saved in fasta format. MEGA software [43] was downloaded, Alignment by ClustalW protein sequence was selected for multiple sequence alignment results, and the model with the lowest BIC score was chosen. Phylogeny was selected to construct the evolutionary tree using the neighbor-joining method at 1000 bootstraps.

### 2.4. Virus-Induced Gene Silencing

Virus-induced gene silencing (VIGS) was performed as previously described [44]. pTRV2-*RhAGL24* vector was constructed through the homologous recombination of a specific *RhAGL24* gene fragment of 150 bp ORF region and 215 bp 3′-untranslated region linked to pTRV2 using *EcoR* I and *BamH* I restriction sites. pTRV1, pTRV2, and pTRV2-*RhAGL24* were transformed into *Agrobacterium tumefaciens* GV3101. Primer sequences are shown in Table S1. pTRV1 and pTRV2 with an absorbance $OD_{600}$ of 1.0 were mixed as control, and pTRV1 and pTRV2-*RhAGL24* were mixed as experimental group and placed in the dark at room temperature for 3 h. More than 100 rooting rose plants were equally incorporated in the two agrobacterium mixtures and placed in a vacuum of −25 kPa for 10 min, and this process was repeated once. Afterward, the plants were washed with deionized water and left in the dark at 8 °C for 3 d before being transplanted to pots with a 1:1 mixture of peat and vermiculite. The plants were then grown at 22 ± 1 °C under 16 h light and 8 h dark photoperiod for 40 d.

### 2.5. Yeast One-Hybrid Assay and Double Luciferase Assay

pGADTA-*RhAGL24* and pAbAi-*proRhARF18* vectors were constructed as previously reported [45]. First, the pAbAi-*proRhARF18* vector was transformed into Y1H yeast strain and coated into SD/-Ura medium. Different Aureobasidin A (AbA) concentrations were prepared and analyzed to identify the one that could inhibit the activity of pAbAi-*proRhARF18*. pGADT7-*RhAGL24* vector was then transformed into pAbAi-*proRhARF18* yeast strain to observe whether it could grow on SD/-Ura/-Leu medium with the above AbA concentration.

In the transactivation experiment of *RhAGL24*, *proRhARF18:LUC* reporter plasmid was constructed by inserting the short fragment of *RhARF18* promoter into the pGreenII 0800-LUC vector upstream of *LUC* gene. The *RhAGL24* sequence was inserted into pGreenII 62-SK vector to construct *pro35S:RhAGL24* plasmid, which was transformed into *A. tumefaciens* GV3101 containing pSoup plasmid. *proRhARF18:LUC* ($OD_{600}$ = 1) and pGreenII 62-SK

empty vector were mixed as the negative control, and *proRhARF18:LUC* and *pro35S:AGL24* vector were used as experimental group. For injection, 1-month-old tobacco was selected. After 3 days, fluorescein was applied to the back of the leaves. A CCD imaging device (CHEMIPROHT 1300B/LND, 16-bit; Roper Scientific) was used to capture LUC images at $-110$ °C. The sequences of relevant primers are shown in Table S1.

### 2.6. Statistical Analysis

Statistical analysis of all data with at least three biological replicates was performed using student's t test in GraphPad Prism 8.0.2 software [46]. Significant differences were denoted by $p < 0.05$ (*), $p < 0.01$ (**) and $p < 0.001$ (***).

### 3. Results

#### 3.1. Phylogenetic Relationship and Conservation of AGL24 Genes

To investigate the molecular regulatory mechanisms of flower development, we statistically analyzed the differentially expressed genes from transcriptome sequencing of rose organ primordia development at different stages. We found that *RchiOBHm_Chr7g0205071* expression was low in the sepal primordia development stage but increased rapidly in the petal and stamen primordia development stage. We identified the gene as *RhAGL24* according to the reference genome [39], designed specific primers for this gene using Primer Premier 5.0 software by referring to the 'Old Blush' genome and cloned *RhAGL24* gene from the cDNA of *R. hybrida* 'Samantha'. *RhAGL24* has an ORF length of 678 bp and encodes 225 amino acids.

To identify the genetic relationship between RhAGL24 and other species, we downloaded 29 AGL24 protein sequences from NCBI and constructed a phylogenetic tree. The results showed that RhAGL24 has the closest relation to strawberry in Rosaceae (Figure 1A). To study the conserved domain of RhAGL24, we used NCBI as a reference and predicted that this gene has two conserved domains, namely, MADS box and K box region. Multisequence comparison with *AGL24* of *A. thaliana* and other species revealed the specific sequence and location of the two conserved domains (Figure 1B). The MADS box family comprises many members and plays an important role in the development of many plant organs, including fruits, roots, stems, leaves, and flowers. As an important member of the MADS family, we further studied *RhAGL24* and its role in plant development.

#### 3.2. Expression Analysis of RhAGL24 in Different Developmental Stages and Organs

AGL24 is involved in flower development in plants, especially during transition from vegetative to reproductive stages in *Arabidopsis*. However, we found significant changes in *RhAGL24* expression during the early rose primordia stage. Early rosa development is divided into eight stages: vegetative development; prophase of transition from vegetative growth to reproductive growth; transition from vegetative growth to reproductive growth; sepal primordia differentiation; petal primordia differentiation; stamen primordia differentiation; pistil primordia differentiation; and completion (stages 1 to 8) (Figure 2A). We found that *RhAGL24* expression increased along the development stage, from being at a low level in the vegetative to sepal primordium differentiation stages to suddenly increasing at the petal primordium differentiation stage and remaining at a high level throughout the subsequent development stages (Figure 2A). In addition, we detected *RhAGL24* expression in various rose organs. The results showed that *RhAGL24* gene had the highest expression in leaf organs such as leaves and sepals, had a high expression in flowers (including petals, stamens, and pistils), and had a low expression in roots and stems (Figure 2B). On the basis of these results, *RhAGL24* is highly expressed in petals, stamens, and pistils during floral organ development and later flower development. Therefore, RhAGL24 may play an important role in the formation and development of petals, stamens, and pistils.

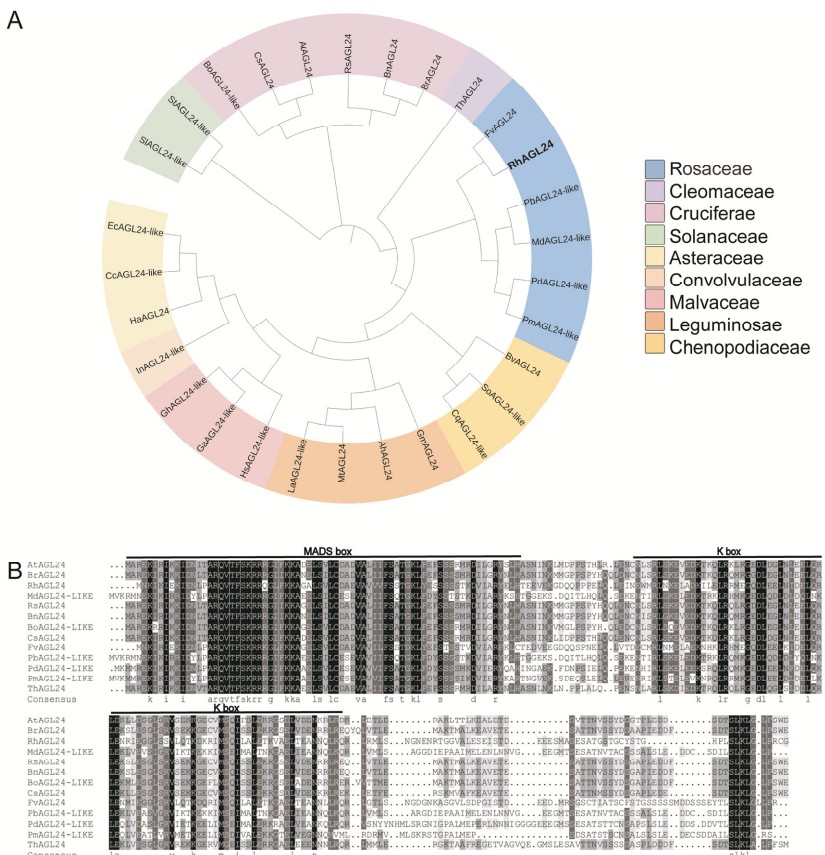

**Figure 1.** Conserved analysis of AGL24 protein sequence. (**A**) Phylogenetic evolutionary trees of AGL24 from different species; (**B**) comparison of multiple sequences of AGL24 proteins from different species. Conservative structures are represented above the alignment, and the amino acids that are the same for all species are represented underneath the alignment. At—*Arabidopsis thaliana*; Br—*Brassica rapa*; Rh—*Rosa hybrida*; Md—*Malus domestica*; Rs—*Raphanus sativus*; Bn—*Brassica napus*; Bo—*Brassica oleracea*; Cs—*Camelina sativa*; Fv—*Fragaria vesca*; Pb—*Pyrus × bretschneideri*; Pd—*Prunus dulcis*; Pm—*Prunus mume*; Th—*Tarenaya hassleriana*.

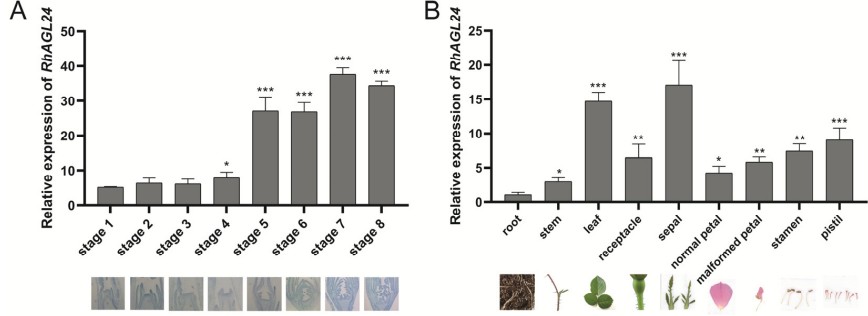

**Figure 2.** Expression analysis of *RhAGL24* gene. (**A**) Relative expression levels of *RhAGL24* at different developmental stages of floral organ primordia. One biological sample consisted of 10 floral buds at any stage. All the values are the means ± SD of three biological replicates. Stage 1—vegetative development; stage 2—prophase of transition from vegetative growth to reproductive growth; stage 3—transition from vegetative growth to reproductive growth; stage 4—sepal primordia differentiation; stage 5—petal primordia differentiation; stage 6—stamen primordia differentiation; stage 7—pistil primordia differentiation; stage 8—completion. (**B**) Relative expression levels of *RhAGL24* in different plant organs. All the values are the means ± SD of three biological replicates. *RhUBI2* was used as an internal control. Asterisks indicate statistically significant differences by Student's t-test (* $p < 0.05$, ** $p < 0.01$ and *** $p < 0.001$).

### 3.3. Silencing RhAGL24 Affected the Number of Petals and Stamens in Rose

To test the role of RhAGL24 in organ development, we silenced *RhAGL24* in rose plants using VIGS technique. The results showed that *RhAGL24* expression was significantly reduced in TRV-*RhAGL24* lines (Figure 3B) and decreased by 60% in TRV-*RhAGL24* plants compared with that in the control. Two changes occurred in the *RhAGL24*-silenced lines: the number of petals increased, and the number of stamens decreased (Figure 3B).

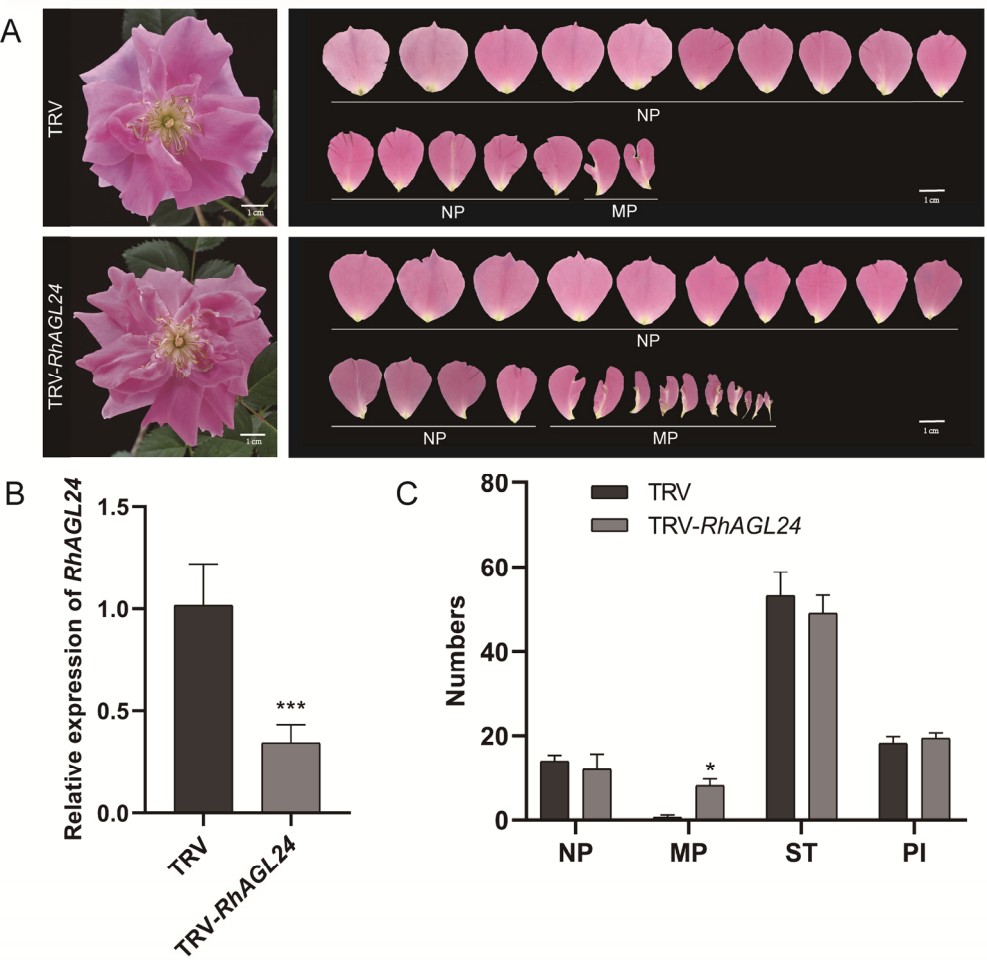

**Figure 3.** Silencing of *RhAGL24* affects the number of petals and stamens. (**A**) Phenotype of TRV and TRV-*RhAGL24* strains. (**B**) *RhAGL24* expression in *RhAGL24*-silenced (TRV-*RhAGL24*) and TRV control plants as determined by qRT-PCR. (**C**) Statistics on the phenotypic data of TRV and TRV-*RhAGL24* strains. NP—normal petal; MP—malformed petal; ST—stamen; PI—pistil. All the values are the means ± SD of seven biological replicates. *RhUBI2* was used as an internal control. Asterisks indicate statistically significant differences by Student's t-test (* $p < 0.05$, *** $p < 0.001$).

According to the morphology of rose petals, we can classify them into normal petals (NPs) and malformed petals (MPs) (Figure 3A). NPs are basically symmetrical on both sides, are relatively intact or have slightly asymmetric left and right sides with concave tips and no stamens. MPs are also asymmetrical on both sides but have distinct stamens. We counted the number of NPs, MPs, stamens, and pistils. Compared with that in the control group, the total petal number increased significantly after *RhAGL24* silencing (Figure 3A). The number of NPs was almost the same but that of MPs increased significantly, the number of stamens decreased but not significantly (Figure 3C). The number of pistils did not change significantly. These results suggested that *RhAGL24* is involved in flower organ development. From a macro point of view, silencing *RhAGL24* increases the number of petals and decreases the number of stamens. However, when the petals are categorized,

*RhAGL24* affects only the number of MPs. Given its high expression in stamen-containing organs (Figure 2B), *RhAGL24* may affect the number of petals and stamens by inhibiting the homologous transformation of stamens to petals.

### 3.4. RhAGL24 Affects the Expression of RhAG and Auxin-Related Genes

Flower development is an extremely complex process and involves many genes, such as *AP1*, *AP2*, *AP3*, *PI*, and *AG*, as indicated in the famous ABCDE model [3]. Multiple genes jointly regulate and form a flower development network. The *AG* of class C gene plays an important role in the development of petals and stamens [47]. In *A. thaliana ag* mutants, almost no stamens exist because they transform into petals. Most of the genes affecting the number of petals and stamens regulate *AG* expression and thus cause phenotypic changes. Therefore, we first detected *RhAG* expression levels in TRV and TRV-*RhAGL24* lines. The results showed that *RhAG* was significantly down-regulated in TRV-*RhAGL24* plants (Figure 4). We detected the similarity between the fragment of RhAGL24 used for VIGS and *RhAG* gene, and ruled out the possibility that this fragment could silence *RhAG*. Therefore, the results show that *RhAGL24* may directly or indirectly affect *RhAG* expression and thus regulate petal and stamen development.

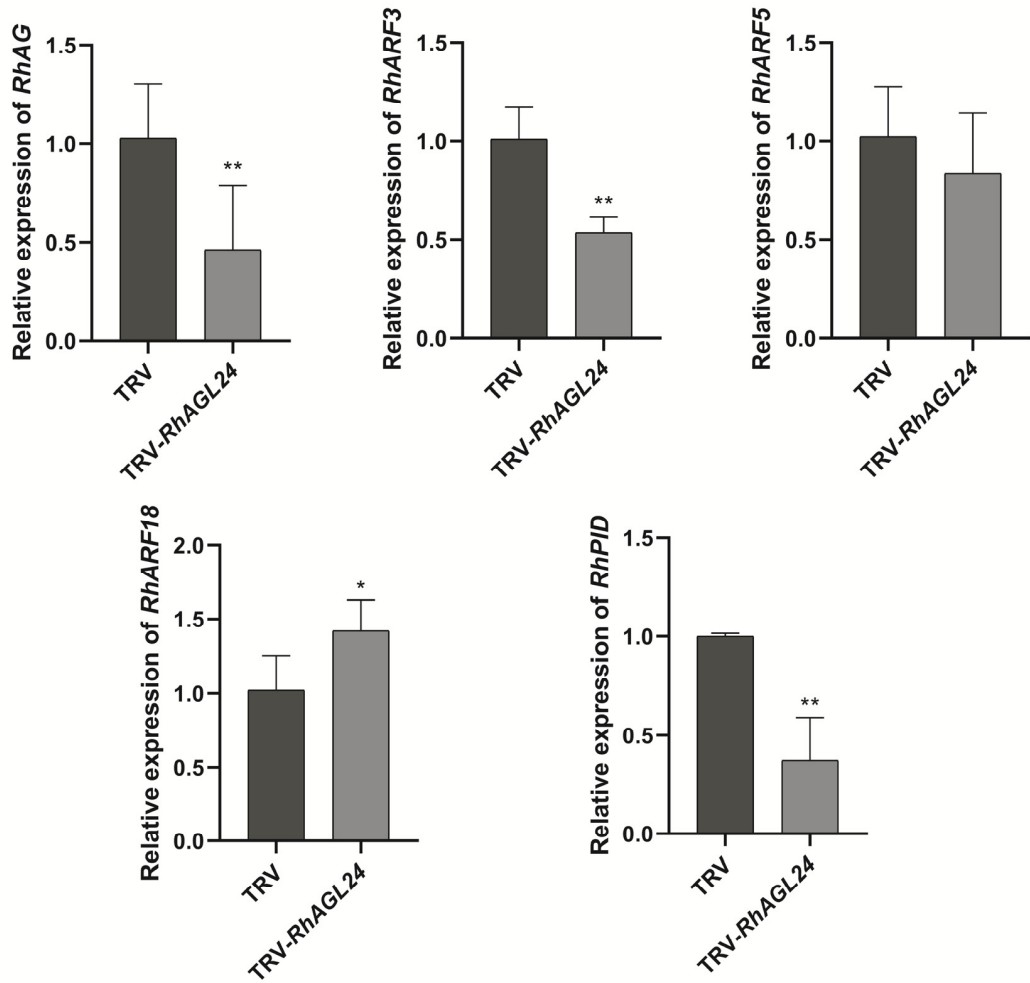

**Figure 4.** Expression of flower organ identity-related genes in *RhAGL24*-silenced plants. All the values are the means ± SD of seven biological replicates. *RhUBI2* was used as an internal control. Asterisks indicate statistically significant differences by Student's t-test (* $p < 0.05$, ** $p < 0.01$).

Plant hormones play an important role in plant development; for example, auxin regulates floral organ development [15,48]. The single and double petal transcriptomes of *Malus* showed significant difference in auxin-related gene expression [28]. ARF3 is

involved in floral meristem determinism, and ARF18 regulates *AG* transcription [19,27]. Therefore, we selected four auxin-related genes (*ARF3*, *ARF5*, *ARF18*, and *PID*) related to flower organ development and detected their expression levels in TRV and TRV-*RhAGL24* lines. The results showed that in *RhAGL24*-silenced plants, *RhARF3* and *RhPID* expression decreased, *RhARF5* expression did not change significantly, and *RhARF18* expression increased (Figure 4). Therefore, we hypothesized that *RhAGL24* changes the number of petals and stamens by affecting the expression of many flower development genes and auxin-related genes and participating in the network of flower organ development.

### 3.5. RhAGL24 Inhibits RhARF18 Transcriptional Activity in Rose

Given that *RhAGL24* is a transcription factor, we predicted a recognizable *cis*-acting element CArG sequence using an online software JASPAR (Figure 5A). We used FIMO to predict the auxin-related gene promoter on the *cis*-acting element. The results showed that only the promoter of *RhARF18* had three CArG sequences; the element in the promoter of the other genes was not predicted (Figure 5B). Hence, we cloned and corrected the promoter sequences of the gene. We applied yeast one-hybrid assay to detect whether *RhAGL24* could bind to the *RhARF18* promoter in vitro. Considering the three predicted binding sites, we cut the promoter into three segments named P1, P2, and P3 (Figure 5B) with one promoter each. Experimental results showed that when the AbA concentration was 300 ng ml$^{-1}$, the RhAGL24 protein could bind to *RhARF18* promoter P1 and activate the reporter gene expression. However, the other two promoters are self-activated and could not be inhibited by AbA (Figure 5C). Therefore, we need to explore other ways to determine whether these two sites are recognized and bound by RhAGL24. These results suggested that RhAGL24 recognizes and binds to the *RhARF18* promoter in yeast and regulates RhARF18 expression at the transcriptional level.

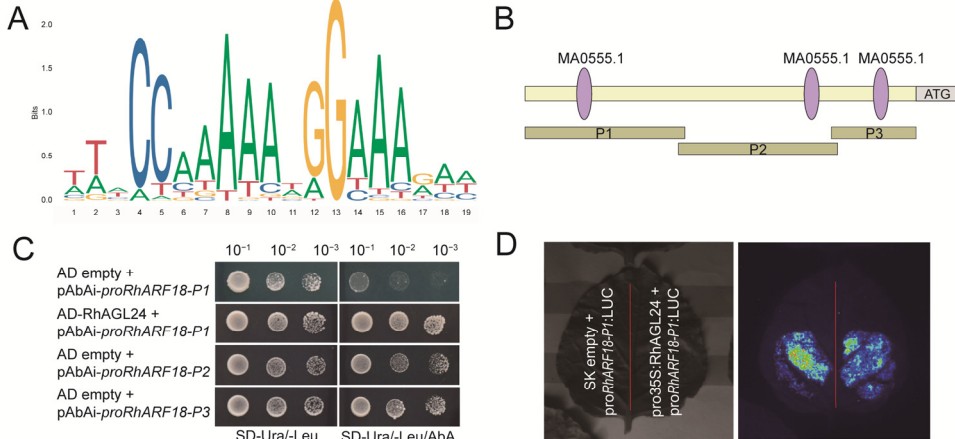

**Figure 5.** Validation of *RhAGL24* downstream genes. (**A**) Prediction of *RhAGL24* identifiable sequences. (**B**) Sequence analysis of *RhARF18* promoter. The *RhARF18* promoter was divided into three segments of P1, P2, and P3, with one predicted *cis*-acting element each. MA0555.1 is the *cis*-acting element of *RhAGL24* in (A). (**C**) Yeast one-hybridization between *RhAGL24* and *RhARF18* promoter; $10^{-1}$, $10^{-2}$, and $10^{-3}$ are the concentrations of diluted yeast. (**D**) Dual luciferase assay of *RhAGL24* and *RhARF18* promoter. The *proRhARF18:LUC* construct was co-infiltrated with *pro35S:RhAGL24* or SK empty vector into *N. benthamiana* leaves. The picture on the left shows the tobacco in the open field, and the picture on the right shows the tobacco in the fluorescent field.

We further verified the above finding by conducting double luciferase assay. The results showed that the fluorescence of RhAGL24 and *RhARF18* promoters simultaneously injected was significantly lower than that of the individual *RhARF18* promoters injected alone (Figure 5D). Therefore, RhAGL24 inhibits the transcriptional activity of

*RhARF18* promoter, thus affecting *RhARF18* expression and changing the number of petals and stamens.

## 4. Discussion

Along with *SVP*, *SOC1*, *LFY*, and other genes, *AGL24* affects the transformation of plants from the vegetative stage to the reproductive stage, thus influencing the flowering time [30,31]. In addition, this gene is associated with inflorescence development and plays a role in inflorescence determinism [49]. However, *AGL24* in floral organ development has rarely been investigated. In this study, we found that *RhAGL24* is highly expressed during petal and stamen primordia development and later flower organ development. Its silencing in rose significantly increases the number of petals, decreases the number of stamens, and enhances stamen petaloidy. Rosaceae plants have many double petals, resulting from stamen petaloidy. *AGL24* expression also significantly differs in the single and double flower transcriptomes of *Malus* and is significantly decreased in double flower [28], a result consistent with the present findings. This work provides a new basis for the double petal phenomenon of Rosaceae and offers new directions and theoretical support for the breeding of new rose varieties.

Plant hormones participate in various plant development, and auxin plays an important role in flower organ development [50]. At present, many auxin-related genes have been studied in relation to flower organ development [9]. Different auxin-related genes are enriched in the above-mentioned *Malus* transcriptome. For example, double-petal plants have increased *ARF18-LIKE* expression and decreased *ARF4-LIKE* expression [28]. We previously found that the number of petals decreased after silencing *RhARF18* in rose and that *RhARF18* could regulate *RhAG* expression [27]. After the mutation of serine threonine kinase *PID*, the petals increase in number and appear as a filiform, the sepals and stamens are reduced, and fusion is exhibited by each flower organ [13,48]. In the present study, we found that auxin-related genes were also changed in pTRV2-*RhAGL24* strains. For example, *RhARF3* and *RhPID* were downregulated, and *RhARF18* was upregulated. This finding suggests that auxin-related genes are involved in the homologous transformation of stamens and petals during flower organ development. In addition, yeast one-hybrid and double luciferase assay showed that RhAGL24 could inhibit regulate *RhARF18*, which in turn could suppress *RhAG* transcription. Therefore, the present study complements the regulatory network of petal and stamen development.

The regulatory network of floral organ development is complex and many of its pathways remain undiscovered. Hence, additional experiments are needed to reveal these pathways.

## 5. Conclusions

*RhAGL24* expression increased rapidly at petal primordium development stage and maintained a high level until the complete differentiation stage. This gene also plays an important role in the later flower organ development. Its silencing increased the number of MPs and decreased the number of stamens, indicating that *RhAGL24* affects stamen petaloidy. In *RhAGL24*-silenced plants, *RhAG* was also downregulated. Moreover, *RhAGL24* inhibited the transcriptional activity of the promoter of *RhARF18*, a regulator of *RhAG*. Our results suggested that RhAGL24-*RhARF18* module regulates stamen petaloidy in rose. This work provides new insights into the function of *AGL24* for plants and serves as a reference for the breeding of new rose varieties.

**Supplementary Materials:** The following supporting information can be downloaded at: https://www.mdpi.com/article/10.3390/horticulturae8050407/s1, Table S1: Primers used for the experiment.

**Author Contributions:** Conceptualization, J.G. and X.S.; methodology, Y.G. and H.R.; software, Z.W.; formal analysis, X.S.; investigation, L.L.; data curation, L.L.; writing—original draft preparation, L.L.; writing—review and editing, X.S., Y.J., N.M. and J.G.; funding acquisition, J.G. and X.S. All authors have read and agreed to the published version of the manuscript.

**Funding:** This research was funded by the National Natural Science Foundation of China, grant number 31902059, Yunnan academician expert workstation, grant number 202105AF150036, Science and Technology Program of Yunnan Province, grant number 202102AE090001.

**Institutional Review Board Statement:** Not applicable.

**Informed Consent Statement:** Not applicable.

**Data Availability Statement:** Not applicable.

**Acknowledgments:** We sincerely thank all colleagues who made suggestions on this manuscript.

**Conflicts of Interest:** The authors declare no conflict of interest.

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
