# Peer review of "RhAGL24 Regulating Auxin-Related Gene RhARF18 Affects Stamen Petaloidy in Rose"

_horticulturae, doi:10.3390/horticulturae8050407_

Round 1

Reviewer 1 Report

In reviewed manuscript, “RhAGL24 regulating auxin related gene RhARF18 affects stamen petaloidy in rose” authors found that RhAGL24 is strongly related to petal and stamen development in rose. The expression level of RhAGL24 elevated in the petal primordium development stage and maintains a elevated level until the complete differentiation stage. Further, silencing of RhAGL24 enhanced the number of malformed petals, while decrease the number of stamens in RhAGL24-silenced plants. Furthermore, authors reported that RhAGL24 could directly bind to the promoter region of RhARF18 (AUXIN RESPONSE FACTORS 18), a regulator of RhAG. Therefore, these findings demonstrated that RhAGL24-RhARF18 regulates stamen petaloidy in rose and provide new insights into the function of AGL24 for plants. The manuscript deals with very interesting and important topic of RhAGL24 in rose. In general, the manuscript represents very big piece of research in logical presentation. Therefore, it might be conditionally accepted with a subject to major revision. However, this major revision means only that there is no necessity to repeat or extend the experiments and their analysis. Instead, authors have to improve their manuscript with many non-clear meaning, inaccuracy and inconsistency, and the authors need to address the following issues before it can be accepted for publication.

  • Introduction Grammatical issues appear to be most prevalent in the introduction, making for very confusing reading. The introduction can be improved by adding more information about auxin and role in flower development.
  • General note: the figures in this section are quite low resolution and difficult to make out. Higher-res versions will be needed for publication for instance, figure 1.
  • Materials and Methods As a general note, the authors cite many different online tools used in the analysis by listing their html addresses. However, most of these tools are based on peer-reviewed publications, and many request that authors cite the publication that the tool is based off. In general most of these website links can (and should) be replaced with appropriate article citations.
  • I would like Authors to provide in the methodology and results the number of replications wherever possible. The same applies to the statistical significance of the results. Please describe statistical methods used in the work in materials and methods.
  • qRT-PCR methodology provided is also very vague and confusing. Please provide more details like what was the calibrator used in the study. I assume the authors have used the control as the calibrator. If so, the authors should not include the control within the bar graph as it represents the fold change between the treated vs control and a fold change of “1” for the ‘control’ doesn’t make any sense. Also, would be good to provide details on what reagents (details of probes used, if any, if SYBR was used then details for that, etc.) and real time PCR machine were used in the current study.
  • The discussion is unnecessarily lengthy and should be more focused based on the results obtained. Please avoid speculation that is not supported by either data you have obtained or available in literature. Results and Discussion’ to make it more concise and avoid repetitions. I leave it to the editor to make the decision.

Reviewer 2 Report

The manuscript by Liu et al. (2022)- 'RhAGL24 regulating auxin related gene RhARF18 affects stamen petaloidy in rose' has merit and the experiments conducted are logical.  I have two minor points that the authors need to address. 

  1. What is the transcript similarity between RhAGL24 (VIGS selected region) and RhAG transcript? Is the silencing directly due to the off-target effects on RhAG by RhAGL24 siRNAs?
  2. Please mention in the legends how many plants (biological replicates) were used in the experiments described in Fig.  2A-B, Fig. 3B-C, and Fig. 4?

Round 2

Reviewer 1 Report

Dear Editor,

Thank you for providing the opportunity to review the revised manuscript. . The manuscript is improved considerably after revision according to the reviewers' comment. Now this study is the suitable contribution to Horticulturae. I recommend the manuscript for publication.

Thank you

With best regards

Reviewer 2 Report

The authors have included my previous suggestions. I have no more comments on the revised manuscript.